# BF_2_-Azadipyrromethene Fluorophores for Intraoperative Vital Structure Identification

**DOI:** 10.3390/molecules28052167

**Published:** 2023-02-25

**Authors:** Cathal Caulfield, Dan Wu, Ian S. Miller, Annette T. Byrne, Pól Mac Aonghusa, Sergiy Zhuk, Lorenzo Cinelli, Elisa Bannone, Jacques Marescaux, Sylvain Gioux, Michele Diana, Taryn L. March, Alexander L. Vahrmeijer, Ronan Cahill, Donal F. O’Shea

**Affiliations:** 1Department of Chemistry, Royal College of Surgeons in Ireland (RCSI), D02 PN40 Dublin 2, Ireland; 2Precision Cancer Medicine Group, Department of Physiology and Medical Physics, Royal College of Surgeons in Ireland (RCSI), D02 PN40 Dublin 2, Ireland; 3National Pre-clinical Imaging Centre (NPIC), Royal College of Surgeons in Ireland (RCSI), D02 PN40 Dublin, Ireland; 4IBM Research-Ireland, Damastown Industrial Estate, Mulhuddart, D02 PN40 Dublin 15, Ireland; 5Research Institute against Digestive Cancer (IRCAD), 67000 Strasbourg, France; 6Department of Gastrointestinal Surgery, San Raffaele Hospital IRCCS, 20132 Milan, Italy; 7Department of Surgery, Istituto Fondazione Poliambulanza, 25124 Brescia, Italy; 8Department of Pancreatic Surgery, Verona University, 37134 Verona, Italy; 9ICube Lab, Photonics Instrumentation for Health, 67400 Strasbourg, France; 10Digestive and Endocrine Surgery, Nouvel Hospital Civil, University of Strasbourg, 67000 Strasbourg, France; 11Department of Clinical Pharmacy and Toxicology, Leiden University Medical Center, 2333 ZA Leiden, The Netherlands; 12Department of Surgery, Leiden University Medical Center, 2333 ZA Leiden, The Netherlands; 13UCD Centre for Precision Surgery, School of Medicine, University College Dublin, D02 PN40 Dublin 4, Ireland; 14Department of Surgery, Mater Misericordiae University Hospital, D02 PN40 Dublin 7, Ireland

**Keywords:** BF_2_-azadipyrromethene, pegylation, NIR-fluorescence, ureter identification, fluorescence guided surgery

## Abstract

A series of mono- and bis-polyethylene glycol (PEG)-substituted BF_2_-azadipyrromethene fluorophores have been synthesized with emissions in the near-infrared region (700–800 nm) for the purpose of fluorescence guided intraoperative imaging; chiefly ureter imaging. The Bis-PEGylation of fluorophores resulted in higher aqueous fluorescence quantum yields, with PEG chain lengths of 2.9 to 4.6 kDa being optimal. Fluorescence ureter identification was possible in a rodent model with the preference for renal excretion notable through comparative fluorescence intensities from the ureters, kidneys and liver. Ureteral identification was also successfully performed in a larger animal porcine model under abdominal surgical conditions. Three tested doses of 0.5, 0.25 and 0.1 mg/kg all successfully identified fluorescent ureters within 20 min of administration which was sustained up to 120 min. 3-D emission heat map imaging allowed the spatial and temporal changes in intensity due to the distinctive peristaltic waves of urine being transferred from the kidneys to the bladder to be identified. As the emission of these fluorophores could be spectrally distinguished from the clinically-used perfusion dye indocyanine green, it is envisaged that their combined use could be a step towards intraoperative colour coding of different tissues.

## 1. Introduction

Dynamic fluorescence imaging has the potential to provide surgical guidance by identifying specific tissues or anatomical structures in real time as an operation proceeds [1]. At present, clinically approved camera hardware exists in a variety of formats, including open, laparoscopic and microscopic, which are capable of capturing high-quality fluorescence images during surgery [2]. Surgical resections can be guided in real-time with these devices due to their inherent ease of use, enabling continuous imaging during surgical procedures [3,4]. As a result, there has been a growing interest in developing fluorophores for in vivo and clinical applications using near-infrared (NIR) light with wavelengths between 700 nm and 1400 nm [5]. Due to low tissue autofluorescence and light attenuation, this wavelength range has been described as the clinical imaging window. This offers the potential of increased personalised surgeries with improved patient outcomes. The benefits can be viewed in two broad categories of ensuring the complete removal of diseased tissue thereby preventing reoccurrences or the prevention of accidental damage to vital structures which cause additional unintended patient complications. One such avoidable complication is accidental ureter damage most commonly encountered in complex lower abdominal surgeries [6,7]. Intraoperative ureteral injuries which are not immediately repaired in situ cause patient morbidity and mortality with a significant increase in time spent in hospital and cost of treatments. Unfortunately, the majority of accidental ureter damage goes unnoticed during the operation causing occurrences of renal failure and sepsis and necessitating further corrective surgery [8,9]. The identification of ureters at the outset of a surgery can be non-trivial due to their anatomical location behind the abdominal cavity often obscured by adipose tissue [6,9]. Cancerous and inflammation tissue or previous surgeries may also hinder their detection. Over half of ureter injuries occur during gynaecological operations with a quarter reported from general surgical procedures. Elevated risks have been reported for robotic or minimally invasive surgeries due to the inability to use the traditional tactile approach [10].

Real-time tissue observations through near-infrared fluorescence is having a growing clinical impact as they do not overly impede on the existing surgical workflow. If applied to ureter preservation, key desirable features would be that the fluorescent agent reach the ureters within minutes after administration, the NIR-wavelengths used for detection are distinguishable from clinically used indocyanine green (ICG) allowing both to be used in parallel (for different tissues), and has no toxicity such that a repeat dose could be used to confirm ureter integrity at the end of the operation. As a cyanine-derived dye, ICG is currently the only NIR fluorophore, with emission spanning 800–900 nm, approved by both the FDA and the European Medicines Agency [11]. As a vascularization assessment tool, it is used most commonly during bowel anastomosis surgeries, where it has been shown to reduce complications associated with the surgery and improve patient outcomes [12,13]. Due to its biliary excretion pathway from hepatocytes into the bile, it is unsuitable for renal excretion imaging [14,15]. Recently, other fluorophore classes with similar spectral wavelengths to ICG have been described as alternatives [16,17,18,19].

In this work, we have investigated the synthesis, physical and photophysical characteristics, and ureter imaging capabilities of NIR-AZA fluorophores **1** as their pegylated derivatives (Figure 1) [20,21]. Therapeutic pegylation is a well-established means to prolong the circulation half-life of a drug molecule but its application to in vivo imaging is relatively unexplored [22,23,24,25,26,27]. It was envisaged that an optimal PEG substitution of a fluorophore could be advantageous for several reasons. It is relatively inexpensive when compared to peptide or antibody conjugates, can increase the fluorescence brightness in aqueous biological media, can restrict the non-specific uptake into tissues and may promote efficient excretion via the renal pathway [22].

To this end, a series of pegylated derivatives of the NIR-AZA fluorophores were envisaged each with different degrees of pegylation (Figure 1). The core fluorophore used was structurally based on the bis-arylmethoxy-substituted NIR-AZA **1a** which has absorbance and emission maxima of 682 and 721 nm, respectively, with a fluorescence quantum yield of 0.35 [28]. Two design approaches were envisaged for PEG attachment to the fluorophore through either a mono or dual covalent connection of the PEG groups which would allow different degrees of pegylation to be explored (Figure 1A,B). 

## 2. Results and Discussion

### 2.1. Synthesis

The strategy adopted for covalent PEG attachment was to functionalize through the *para*-phenolic oxygen substituents of the aryl rings at the *α*-pyrrole positions (Figure 1). This required the synthesis of the bis-phenol **1b** and mono-phenol-monomethoxy **2** as advanced intermediates (Figure 1). The synthesis of **1b** has been previously reported in three steps from commercial starting materials 4-hydroxyacetophenone and nitromethane [29]. Two routes were developed to **2**, the first of which required the reaction of **1b** with iodomethane in DMSO with cesium fluoride as base, which gave the mono-alkylated product in 83% yield (Figure 1). An alternative five-step route which avoided the use of methyl iodide started from 4-methoxyacetophenone and 4-hydroxyacetophenone providing **2** in an overall 8% yield for the route (Appendix A) [30].

The conversion of **1b** and **2** to their bis- or mono-*N*-hydroxysuccinimide esters was achievable in three steps through first phenolic *O*-alkylation with *tert*-butyl bromoacetate (Figure 2). These nucleophilic substitutions were carried out under basic conditions in anhydrous DMSO at 40 °C or THF under reflux with monitoring by thin layer chromatography. Upon completion, work up involved silica gel chromatography purifications with **3** and **5** isolated in 84 and 86% yield, respectively. The NMR and mass spectrometry analysis confirmed the structures of both products. Next, *tert*-butyl ester cleavage of both were achieved with trifluoroacetic acid at room temperature in CH_2_Cl_2_ over 3 h. The carboxylic acid products **3a** and **5a** precipitated during the reaction allowing their isolation by filtration, following which they were washed with CH_2_Cl_2_ and dried under vacuum (Appendix A). Rather than attempt a direct acid/amine-PEG coupling it was preferential to first make and isolate the mono- and bis-substituted-activated esters **4** and **6**. Room temperature reaction of the mono- and bis-carboxylic acids with *N*-hydroxysuccinimide in anhydrous DMSO in the presence of a carbodiimide coupling reagent EDCI showed complete conversion to esters **4** and **6** after 2 h as judged by analytical HPLC (Figure 2). Both products could be isolated following the addition of 0.5 M HCl to the reaction mixture and extraction into CH_2_Cl_2_. The organic layer was washed with H_2_O and brine, dried over Na_2_SO_4_, filtered and reduced to dryness providing the coupling-enabled fluorochromes to be isolated as solids. 

The fluorochromes were then reacted with amino-PEG polymers with molecular weights of 0.55, 4.6 or 9 KDa for **4** and 0.55, 2.9 or 4.6 KDa for **6** (Figure 3) [31]. Reactions were carried out in DMSO at rt with conversions monitored using analytical HPLC with all reactions reaching completion within 60 min. Reactions for **4** and **6** were carried out under identical conditions, though the equivalence of H_2_N-PEG-OH used was 1.05 for the former and 2.1 for the latter. Purifications entailed the removal of the solvent, redissolving in water and chromatography using a Sephadex G-25 column. For photophysical measurements and in vivo studies, the products were further purified by reverse phase semi-prep chromatography to yield PEGylated fluorophores **7**–**12** in greater than 95% purity (Figure 3). 

The conversion of *tert*-butyl ester **3** to mono-pegylated NIR-AZA fluorophores **7**–**9** can be easily followed by ^1^H NMR spectroscopy through the change in chemical shift of the methylene protons which is unobstructed by the large PEG proton signals further up-field (Figure 2). For example, a chemical shift of 4.59 ppm is observed for *tert*-butyl ester **3** going to 5.04 ppm for *N*-succinimidyl ester **4** and returning to a similar chemical shift of 4.58 ppm for PEG-amide **7**. Contrastingly, the methoxy methyl ^1^H NMR signal remained relatively constant at 3.85–3.9 ppm for **3**, **4** and **7**. The covalent attachment of the PEG groups was further confirmed by MALDI Q-TOF mass spectrometry with the expected mass ranges obtained.

### 2.2. Photophysical Properties of ***7***–***12***

Next, the photophysical characteristics of each pegylated fluorophore were examined in acetonitrile and water. The properties of **7**–**12** in acetonitrile were very similar with λ_max_ of absorbances at 675 (±5) nm, emissions at 714 (±3) nm and quantum yields between 0.26 and 0.32 (Table 1). As aqueous solutions, both absorption and emission maxima were slightly red shifted at 685 (±3) and 723 (±2) nm, respectively. However, when quantum yields were compared as aqueous solutions, interesting differences were revealed between the mono **7**–**9** and dual **10**–**12** PEG-substituted derivatives of similar molecular weights. For example, both **9** and **12** have comparable overall PEG mass (i.e., 1 × 9 kDa for **9** and 2 × 4.6 kDa for **12**), yet the quantum yield of **12** is four-fold higher than that of **9** (Table 1, entries 3 and 6). A similar trend was observed for mono-PEG **8** and dual-PEG **11** with quantum yields of 0.11 and 0.4, respectively (entries 2 and 5). The higher quantum yields for the dual substituted derivatives could be attributable to better shielding of the fluorophore from the bulk water by having two points of PEG attachment rather than one. A comparison of the absorption spectra of **8** with **11** and **9** with **12** showed that the mono-substituted compounds had slightly broadened spectra with full width at half maximum (fwhm) values of 109 versus 94 nm and 114 versus 93 nm, respectively (Figure 3A,B). While only subtle differences were noted in the absorbance, this did have a larger impact in the fluorescence intensity which was markedly higher for the dual substituted derivatives **11** and **12**. Clearly, PEG molecular weight alone is not the sole determinant of aqueous quantum yields with the double substitution providing a more favourable solvated micro-environment for the fluorophores, resulting in higher quantum yields. The impact of water was confirmed by comparing absorption and emission spectra of **12** in water with that of acetonitrile (Figure 3C). This showed that absorbance was sharper in acetonitrile (fwhm 75 nm) with the quantum yield 2.5 times higher than water indicating that pegylation is a good strategy to impart aqueous solubility for fluorophores but the number and position of the PEG groups may be required to optimize quantum yield.

Assessment of the most promising fluorophores **11** and **12** using clinical imaging equipment was next carried out. Polytetrafluoroethylene (PTFE) tubing of 2 mm external diameter was filled with PBS solutions of **11** or **12** at concentrations of 5.0, 1.0 or 0.5 µM. Typically, a human ureter has a diameter of 4–6 mm and may be masked by adipose tissue, so prior to imaging, tubes were horizontally placed in a vessel and covered with either water or 1% intralipid solution to a depth of 1 cm to simulate the effect of light scattering through tissue [32]. Images were acquired using a Quest Spectrum clinical instrument fitted with an open camera positioned 30 cm from the tubes and excitation wavelength set at 680 nm and collection filters from 700 to 830 nm [3,33]. Encouragingly, in all cases the linear fluorescence from within the tubes could be detected with intensity decreasing with lower concentrations. As would be expected, fluorescence scattering was noted for tubes immersed in intralipid solution though this did not inhibit identifying their positions even at the lowest concentration of 0.5 µM (Figure 4).

### 2.3. Ureter Identification in Rodent Model

As a preliminary screening of in vivo brightness, equal 1 mg/kg concentrations of fluorophores **8**, **9**, **11**, and **12** were administered by tail vein injections and accumulation of fluorophores in the bladder was estimated by comparison of fluorescence intensity at 60 min through the intact skin and tissue. It was found that while all fluorophores accumulated in the bladder **11** and **12** performed better than **8** and **9** based upon averaged fluorescence intensity measurements. The relative bladder emission intensity, taken 1 h post administration, for the four fluorophores **8**:**9**:**11**:**12** was 1.3:1:1.6:1.9 (Appendix A). While there was only a minor difference between **11** and **12**, the marginally better **12** was chosen for more advanced in vivo studies.

To gain a first estimate range of concentrations of **12** useful for imaging, a dose de-escalation study was undertaken using 0.2, 0.1 and 0.05 mg/kg of **12**. It was confirmed that each dose gave rise to a sufficient accumulation of **12** in the bladder at 60 min such that fluorescence was clearly detectable through the intact tissue and skin (Appendix A). Urine samples were taken for each dosage and the presence of intact **12** was confirmed by fluorescence spectroscopy and HPLC (Figure 5). While fluorescence intensities fluctuated due to variable urine volumes, in each case the expected emission maximum (721 nm) and HPLC retention time was comparable to a standard analysis of **12**. Using a standard concentration versus fluorescence intensity of **12** for comparison, it could be estimated that 30% of **12** had been excreted via the renal pathway by this time (Appendix A).

Next, surgical exposure of the lower abdomen was carried out prior to fluorophore administration such that direct observation of the ureters and other abdominal organs was possible. Fluorophore **12** was selected for this study being administered at a dose of 0.5 mg/kg with imaging continued for 90 min. Immediately following the tail vein injection, the blood vessel fluoresced with the kidneys also becoming fluorescent within seconds which was sustained throughout the 90 min (Appendix A). In each animal the ureters were identifiable within 20 min post administration and this maintained throughout the 90 min imaging period. A representative image (displayed both as fluorescence pseudo coloured green overlaid on the white light image and as fluorescence alone in black and white) is shown in Figure 6 panel A which was recorded after 35 min. In this image, both ureters (red arrows) are visible as is the highly emissive bladder (asterisk) (see Appendix A). A clear predisposition of fluorescence from the kidneys relative to the liver and other organs existed throughout, confirming the preference for renal clearance of **12**. A representative example of this is shown in Figure 6 panel B which was recorded 50 min post administration. In this case the animal was positioned to allow a clearer view of one ureter (red arrow), a kidney (circle), spleen (triangle) and liver (square) such that a relative fluorescence intensity quantification could be carried out for each organ (Figure 6, panel B, left). An averaged pixel intensity was determined for four regions of interest of the ureter, kidney, spleen and liver as indicated by the yellow boxes (panel B, right). The emission ranged from highest to lowest for the ureter >> kidney >> liver > spleen with ratio intensity values of 4.8:2.8:1.1:1, respectively (Figure 6B, Appendix A for pixel analysis data).

### 2.4. Toxicity Study of ***12***

A toxicology study of **12** was conducted using six weeks old male Sprague–Dawley rats weighing between 130 and 240 g. The objective of this study was to evaluate **12** using two different dosages administered by an intravenous (bolus) tail vein injection. Due to the fact that this was a preclinical acute toxicity study, high doses were used since the results would provide valuable information about this compound. The doses tested were (i) seven doses of 2 mg/kg administered on seven consecutive days amounting to a total of 14 mg/kg and (ii) a single injection of 10 mg/kg. PBS solutions of **12** were prepared at concentrations of 0.4 or 2 mg/mL, with PBS used as a control. Blood samples were acquired on day 8 and following euthanasia, organs were collected and weighed. No unscheduled deaths occurred during the study and no test item-related clinical signs were observed during the study. In comparison to control animals, there was no effect on body weight during the course of the study (Figure 7). No organ weight changes occurred, at any of the doses tested, that were considered to be related to the administration of **12** (Appendix A).

The haematology parameters measured included erythrocyte count, hemoglobin, thrombocyte count, leucocyte count, differential white cell count and reticulocyte count. Blood chemistry for sodium, potassium, calcium, chloride, inorganic phosphorus, glucose, urea, creatinine, total bilirubin, total protein, albumin, albumin/globulin ratio, total cholesterol, triglycerides, alkaline phosphatase, alanine aminotransferase, aspartate aminotransferase and bile acids were determined. No statistically significant differences from controls for blood biochemistry parameters were observed during the study with for example, alanine transaminase (ALT), aspartate transaminase (AST), creatinine and urea levels comparable for both treated and control animal groups (Appendix A). No statistically significant haematology differences were observed for white blood cells, neutrophils, lymphocytes, or monocytes relative to controls (Table 2). 

A complete *post-mortem* examination was performed on all animals, including examination of the external surfaces, organs and tissues. No adverse macroscopic or microscopic findings were observed, and no histologic changes were detected in the liver tissue. At the injection sites, microscopic findings were observed with similar incidences and severities as those in controls and were considered to be related to the injection procedure. The study outcome recorded a No Observable Adverse Effect Level (NOAEL) at 10 mg/kg/day for **12**. 

### 2.5. Porcine Ureter Imaging

As a next step, ureteral identification with **12** was performed in porcine models under abdominal surgical conditions. Three experimental quantities were selected as 0.5, 0.25 and 0.1 mg/kg in order to identify a range of potential clinical doses (*n* = 2 for each dose). It is worth noting that as **12** incorporates a dual 4.6 KDa PEG substituent, only 6.3% of the total molecular weight arises from the fluorophore itself. With respect to just the fluorophore component of **12**, these doses would be in the range of 0.028 to 0.0056 mg/kg. Images with **12** were acquired using a pre-clinical open-camera system (Perkin Elmer Solaris) by excitation at 660 nm and fluorescence collection between 700–800 nm while simultaneously capturing white light video [34]. This instrument also had the ability to image ICG using a 790 nm excitation and collection between 800–900 nm which allowed confirmation that both **12** and ICG could be visualised independently of each other. The goals were to identify the earliest time point of emissive ureters, establish ureter fluorescence longevity and confirm that **12** and ICG could be imaged independently of each other. Experimentally, images were recorded post administration at 10, 20, 40, 80 and 120 min (Figure 8, for other representative images see Appendix A). 

Encouragingly, for all concentrations tested, ureters could be identified from the fluorescence image by 20 min post-administration. For the lowest dose of 0.1 mg/kg, ureteral fluorescence was still observable at 80 min and for the two higher doses the fluorescence remained strong until the end of the experiment at 120 min. An ICG reference phantom card, which was included in the imaging field of view (FOV), was non-emissive during this imaging, confirming wavelength fidelity. At this stage, administration of ICG (0.5 mg/kg) was carried out and ureters re-examined to check the emission from **12** (excitation 660 nm) was recordable without interference from ICG, while ICG emission (excitation 790 nm) was notable from the phantom card and other regions of the bowel (Figure 9). This confirmed the clinical potential for both **12** and ICG to be utilised simultaneously to colour code different tissues. 

### 2.6. 3D Fluorescence Intensity Heat Map

The passage of urine from the kidney to the bladder is controlled by a ureteral peristalsis mechanism. Ureteral peristalsis is the wave of muscle contraction that pumps urine, in pulses, through the ureter from the kidney into the bladder. This means that as long as a fluorophore remains within the urine and does not enter the cells comprising the ureter, then the emission intensity emanating from the ureter will rise and fall in response to the ureteral peristaltic phases [35]. This offers an opportunity to distinguish the ureter from any background fluorescence as it would not have this repetitive pulsed modulation of intensity. We have previously developed and reported on a 3-D emission intensity heat map display which could be used to record these dynamic changes in intensity signal [36,37]. To this end, a split display was adopted with normal video and fluorescence overlay pseudo coloured as white on the lower level and the fluorescence intensity map above. Fluorescence is displayed as a heat map from blue (no emission) to red (high emission) and with a normalized intensity scale from 0 to 1. This allows relative brightness to be compared for an entire field of view making spatial and temporal changes in intensity obvious. A representative example recorded over 5 s is shown in Figure 10, in which the intensity rise and fall can be tracked, due to a single peristaltic wave, in the heat map time view (Appendix A). It is hoped that such recording of these intermittent visualizations coupled to tissue recognition software would allow the ureter location to be identifiable for the entire surgery.

## 3. Experimental Section

### 3.1. General

All reagents were used as received without further purification. Air-sensitive reagents were reacted in oven-dried glassware under nitrogen using syringe-septum cap techniques. All solvents were purified and degassed prior to use. Reactions were monitored using (i) thin-layer chromatography techniques (0.25 mm silica gel-coated aluminium plates (60 Merck F254) using 254 nm UV light for visualisation), (ii) reverse phase chromatography on a HPLC (Shimadzu) equipped with an analytical (YMC-Triart Phenyl, 4.6 × 150 mm I.D. S-5 µm) column, eluting with acetonitrile/water, and (iii) ^1^H NMR monitoring. If required, products were further purified using (i) flash-column chromatography techniques with Merck silica gel 60 under pressure or (ii) reverse phase semi-preparative HPLC using a semi-preparative (YMC-Triart Phenyl, 10 × 150 mm I.D. S-5 µm, 12nm) column. Products were analysed by ^1^H NMR, ^13^C NMR and ^19^F NMR spectra recorded at 400 MHz, 101 MHz and 376 MHz, respectively, at rt and calibrated using residual non-deuterated solvent as an internal reference. Chemical shifts are reported in parts-per-million (ppm). Further analysis by reverse phase chromatography was carried out for final products using an HPLC (Shimadzu) as described above for reaction monitoring. ESI mass spectra were acquired using Waters LCT Classic in positive and negative modes as required. MALDI-TOF spectra were acquired using the Bruker ‘autoflex maX’ system with the MALDI Imaging Platform. All absorbance spectra were recorded with a Varian Cary 50 scan UV-visible spectrophotometer and fluorescence spectra were recorded with a Varian Cary eclipse fluorescence spectrophotometer. Organic solvents for absorbance and fluorescence experiments were of HPLC quality and Millipore filter HPLC grade water was used. The clinical Quest Spectrum Platform Imaging system (open camera) was used for in vivo rodent studies and the pre-clinical Perkin Elmer Solaris open-camera system was used for in vivo porcine ureter imaging. Compound **1b** was synthesized as per literature procedures [29].

### 3.2. Synthesis and Characterization

*BF_2_ Chelate of 4-(2-((5-(4-methoxyphenyl)-3-phenyl-1H-pyrrol-2-yl)imino)-3-phenyl-2H-pyrrol-5-yl)phenol,***(2)**. Compound **1b** (200 mg, 0.38mmol) and CsF (287 mg, 1.89 mmol) were dissolved in anhydrous DMSO (4.2 mL) under an N_2_ atmosphere. The mixture was stirred at 30 °C for 20 min. Iodomethane (35 µL, 0.57 mmol) was added to the dark solution and the mixture was stirred at 30 °C for 20 min. The solution was partitioned between ethyl acetate (100 mL) and phosphate buffered saline 10 × (100 mL) and the organic phase was washed with acidic water (2 × 100 mL), brine (100 mL), dried over anhydrous Na_2_SO_4_, filtered and evaporated to dryness. The crude was purified by silica gel column chromatography eluting with CH_2_Cl_2_ first to obtain the unwanted bis-alkylated product (15 mg, 7%), then CH_2_Cl_2_/MeOH 99:1 to obtain the desired product **2** as a red metallic solid (171 mg, 83%). ^1^H NMR (400 MHz, DMSO-*d6*) δ: 8.19-8.11 (m, 8H, Ar*H*), 7.63 (s, 1H, Ar*H*), 7.57-7.43 (m, 7H, Ar*H*), 7.14 (d, *J* = 9.1 Hz, 2H, Ar*H*), 6.95 (d, *J* = 8.9 Hz, 2H, Ar*H*), 3.89 (s, 3H, OCH_3_) ppm. ^13^C NMR (100 MHz, DMSO-*d6*) δ: 161.7, 161.4, 158.5, 156.1, 144.8, 143.9, 142.3, 141.1, 132.4, 132.0, 131.7, 131.5, 129.6, 129.3, 129.1, 129.0, 128.7, 128.6, 123.5, 121.4, 120.1, 119.1, 116.0, 114.4, 55.6 ppm. mp 198–200 °C. HRMS (ES): *m/z* calcd for C_33_H_23_BF_2_N_3_O_2_ [M-H]^−^: 542.1851; found: 542.1856. IR (KBr disk): 1427, 1264 cm^−1^.

*tert-Butyl-2-(4-(5,5-difluoro-7-(4-methoxyphenyl)-1,9-diphenyl-5H-4λ^4^,5λ^4^-dipyrrolo[1,2-c:2′,1′-f][1–3,5]triazaborinin-3-yl)phenoxy)acetate* (**3**). A solution of **2** (146.7 mg, 0.27 mmol, 1.0 eq.) and CsF (168.4 mg, 1.108 mmol, 4.0 eq.) was stirred in anhydrous DMSO (8 mL) and treated with *tert*-butylbromoacetate (0.07 mL, 0.443 mmol, 1.6 eq.) under N_2_. The reaction was heated at 40 °C for 2 h. The mixture was cooled in an ice bath to 0 °C, saturated aqueous NH_4_Cl (40 mL) was added and the product extracted with ethyl acetate (2 × 40 mL). The combined organic layers were washed with brine (80 mL), dried over anhydrous Na_2_SO_4_, filtered and the solvent removed by rotary evaporation. The crude residue was purified by flash silica gel chromatography using CH_2_Cl_2_ as eluent affording a red metallic solid **3** (153.1 mg, 84%), mp = 153–155 °C. ^1^H NMR (400 MHz, CDCl_3_) δ: 8.13–8.02 (m, 8H), 7.50–7.38 (m, 6H), 7.05 (s, 1H), 7.04–6.98 (m, 6H), 4.59 (s, 2H), 3.89 (s, 3H), 1.51 (s, 9H) ppm; ^13^C NMR (101 MHz, CDCl_3_) δ: 167.7, 162.2, 160.2, 158.7, 158.3, 157.8, 145.7, 145.3, 143.6, 143.2, 132.6, 132.6, 131.9, 131.7, 129.4, 129.4, 129.4, 128.7, 125.2, 124.2, 119.0, 118.7, 116.0, 114.9, 114.4, 82.9, 65.8, 55.6, 28.2 ppm; ^19^F NMR (376 MHz, CDCl_3_) δ: −131.87 (q) ppm. HRMS (ESI^+^): *m/z* calcd. for C_39_H_34_BF_2_N_3_NaO_4_ [M + Na]^+^ 680.2509; found 680.2511.

*4-((5,5-difluoro-7-(4-methoxyphenyl)-1,9-diphenyl-5H-4λ^4^,5λ^4^-dipyrrolo[1,2-c:2′,1′-f][1–3,5]triazaborinin-3-yl)phenoxy)methane-1-carboxylic acid* (**3a**). Compound **3** (128.1 mg, 0.195 mmol, 1.0 eq.) was dissolved in anhydrous CH_2_Cl_2_ (5 mL) and trifluoroacetic acid (1.27 mL, 16.6 mmol, 85.0 eq.) was added dropwise over 1 min with stirring under N_2_. The reaction was stirred for 3 h at rt, following which the solvent was removed by rotary evaporation. CH_2_Cl_2_ (3 mL) was added to the crude residue and the mixture sonicated for 2 min, filtered and washed with cold CH_2_Cl_2_ (2 × 10 mL) which yielded the product as a red metallic solid **3a** (93.8 mg, 80%), mp = 193–195 °C. ^1^H NMR (400 MHz, CDCl_3_) δ: 8.14–8.02 (m, 8H), 7.50–7.40 (m, 6H), 7.09–6.97 (m, 6H), 4.75 (s, 2H), 3.90 (s, 3H) ppm (OH not observed); ^19^F NMR (376 MHz, CDCl_3_) δ: −131.88 (q, *J* = 31.2 Hz) ppm. HRMS (ESI^−^): *m/z* calcd. for C_35_H_25_BF_2_N_3_O_4_ [M − H]^−^ 600.1918; found 600.1929.

*N*-Succinimidyl-2-((5,5-difluoro-7-(4-methoxyphenyl)-1,9-diphenyl-5H-4λ^4^,5λ^4^-dipyrrolo[1,2-c:2′,1′-f][1–3,5]triazaborinin-3-yl)phenoxy)acetate (**4**). Compound **3a** (50.0 mg, 0.083 mmol) was dissolved in anhydrous DMSO (1.25 mL) and *N*-hydroxysuccinimide (28.7 mg, 0.249 mmol) and *N*-(3-dimethylaminopropyl)-*N’*-ethylcarbodiimide hydrochloride (31.8 mg, 0.166 mmol) were added under N_2_. The reaction was stirred for 2 h at rt, following which the reaction was partitioned between CH_2_Cl_2_ (15 mL) and 0.5 M HCl (15 mL). The organic phase was extracted and washed with H_2_O (15 mL) and brine (15 mL), dried over Na_2_SO_4_, filtered and the solvent removed by rotary evaporation (water bath set to 30 °C). This afforded a green solid product **4** (48.1 mg, 83%) which was brought forward without further purification, mp = 153–155 °C. ^1^H NMR (400 MHz, CDCl_3_) δ: 8.13–8.02 (m, 8H), 7.48–7.37 (m, 6H), 7.07 (s, 2H), 7.04 (d, *J* = 6.1 Hz, 2H), 7.00 (d, *J* = 3.8 Hz, 2H), 5.04 (s, 2H), 3.89 (s, 3H), 2.87 (s, 4H) ppm; ^13^C NMR (101 MHz, CDCl_3_) δ: 63.3, 55.6, 25.7 ppm; ^13^C NMR (101 MHz, CDCl_3_) δ: 168.7, 164.4, 162.3, 159.1, 157.1, 145.9, 145.1, 144.0, 143.0, 132.7, 132.4, 131.9, 131.7, 129.5, 129.5, 129.4, 129.3, 128.7, 128.7, 126.2, 124.0, 119.2, 118.5, 115.0, 114.5, 63.3, 55.6, 25.7 ppm; ^19^F NMR (376 MHz, CDCl_3_) δ: −131.82 (q, *J* = 21.7 Hz) ppm. HRMS (ESI^+^): *m/z* calcd. for C_39_H_29_BF_2_N_4_NaO_6_ [M + Na]^+^ 721.2047; found 721.2048.

*Di-tert-butyl 2,2’-(((5,5-difluoro-1,9-diphenyl-5H-4λ^4^,5λ^4^-dipyrrolo [1,2-c:2′,1′-f] [1–3,5]triazaborinine-3,7-diyl)bis(4,1-phenylene))bis(oxy))diacetate* (**5**) [31]. A solution of **1b** (160.0 mg, 0.302 mmol) and NaH (60% oil dispersion)(49.0 mg, 2.043 mmol) was stirred in anhydrous THF (15 mL) and treated with *tert*-butylbromoacetate (0.178 mL, 1.202 mmol) at 0 °C under N_2_. The reaction was warmed to rt and stirred under reflux for 3 h with TLC monitoring. The mixture was cooled in an ice bath to 0 °C, saturated aqueous NH_4_Cl (15 mL) was added and the product extracted with ethyl acetate (2 × 20 mL). The combined organic layers were washed with brine (20 mL), dried over anhydrous Na_2_SO_4_, filtered and the solvent removed by rotary evaporation. The crude residue was purified by flash silica gel chromatography using CH_2_Cl_2_ as eluent affording a red metallic solid **5** (167.2 mg, 73%). ^1^H NMR (400 MHz, CDCl_3_) δ: 8.07 (t, *J* = 8.1 Hz, 8H), 7.49–7.38 (m, 6H), 7.05–6.97 (m, 6H), 4.59 (s, 4H), 1.51 (s, 18H); ^13^C NMR (101 MHz, CDCl_3_) δ: ppm; ^13^C NMR (400 MHz, CDCl_3_) δ: 167.7, 160.3, 158.3, 145.5, 143.5, 132.6, 131.8, 129.4, 129.4 128.7, 125.1, 118.8, 115.0 ppm; ^19^F NMR (376 MHz, CDCl_3_) δ: −131.89 (q) ppm. ESI [M + H]^+^: calcd for C_44_H_43_BF_2_N_3_O_6_^+^ 758.6; found 758.7.

2,2′-((5,5-difluoro-1,9-diphenyl-5*H*-4λ^4^,5λ^4^-dipyrrolo[1,2-*c*:2′,1′-*f*][1–3,5]triazaborinin-3,7-diyl)bis(4,1-phenylene))bis(oxy))dicarboxylic acid (**5a**) [31]. Compound **5** (500 mg, 0.660 mmol) was dissolved in anhydrous CH_2_Cl_2_ (45 mL) and trifluoroacetic acid (5.0 mL, 66 mmol) was added dropwise over 1 min with stirring under N_2_. The reaction was stirred for 4 h at rt with HPLC monitoring, following which the solvent was removed by rotary evaporation. CH_2_Cl_2_ (3 mL) was added to the crude residue and the mixture sonicated for 2 min, filtered and washed with cold CH_2_Cl_2_ (2 × 5 mL) which yielded the product as a red metallic solid (365.2 mg, 85%). ^1^H NMR (400 MHz, DMSO) δ: 8.16 (d, *J* = 8.0 Hz, 8H), 7.60 (s, 2H), 7.54 (t, *J* = 7.3 Hz, 4H), 7.50–7.44 (m, 2H), 7.13 (d, *J* = 8.8 Hz, 4H), 4.85 (s, 4H) ppm (OH not observed); ^19^F NMR (376 MHz, CDCl_3_) δ: −130.27 (q, *J* = 32.3 Hz) ppm. ESI [M − H]^−^: calcd for C_36_H_25_BF_2_N_3_O_6_^−^ 644.4; found 643.7.

*Bis-(N-succinimidyl)-2,2′-(((5,5-difluoro-1,9-diphenyl-5H-4λ^4^,5λ^4^-dipyrrolo[1,2-c:2′,1′-f][1–3,5]triazaborinine-3,7-diyl)bis(4,1-phenylene))bis(oxy))diacetate* (**6**) [31]. Compound **5a** (35.0 mg, 0.054 mmol) from the previous step was dissolved in anhydrous DMSO (1.0 mL) and *N*-hydroxysuccinimide (62.4 mg, 0.542 mmol) and *N*-(3-dimethylaminopropyl)-*N’*-ethylcarbodiimide hydrochloride (41.6 mg, 0.217 mmol) were added under N_2_. The reaction was stirred for 3.5 h at rt, following which the reaction was partitioned between CH_2_Cl_2_ (10 mL) and 0.5 M HCl (10 mL). The organic phase was extracted and washed with H_2_O (10 mL) and brine (10 mL), dried over Na_2_SO_4_, filtered and the solvent removed by rotary evaporation (water bath set to 30 °C). This afforded a green solid product **6** (37.4 mg, 83%) which was brought forward without further purification. ^1^H NMR (400 MHz, CDCl_3_) δ: 8.07 (dd, *J* = 14.9, 7.9 Hz, 8H), 7.49–7.38 (m, 6H), 7.07 (d, *J* = 8.9 Hz, 4H), 7.02 (s, 2H), 5.05 (s, 4H), 2.85 (s, 8H) ppm; ^19^F NMR (376 MHz, CDCl_3_) δ: −130.18 (q, *J* = 32.0 Hz) ppm.

*Pegylated fluorophore* (**7**). Compound **4** (54.0 mg, 0.077 mmol, 1.0 eq.) and *O*-(2-aminoethyl) polyethylene glycol 0.55 kDa (45.2 mg, 0.083 mmol) were dissolved in anhydrous DMSO (2.0 mL) and stirred at rt for 1 h under a N_2_ atmosphere. The solvent was removed by lyophilization and the crude product dissolved in H_2_O (20 mL) and extracted with CH_2_Cl_2_ (3 × 50 mL). The organic layers were combined, washed with aqueous HCl (pH 5, 20 mL), H_2_O (20 mL), brine (20 mL), dried over anhydrous Na_2_SO_4_, filtered and evaporated to dryness. The green solid was dissolved in H_2_O (2.0 mL) and was first purified by aqueous size exclusion chromatography (Sephadex G-25). The combined fractions were dried by lyophilization which yielded a green solid (85.2 mg, 98% yield). For in vivo studies, this material was further purified as follows: diluted in ACN:H_2_O (1.5 mL; 50:50), filtered through a PTFE 0.45 µM syringe filter and purified by reverse phase semi-prep chromatography (YMC-Triart Phenyl, 10 × 150 mm I.D. S-5 µm, 12 nm; injection volumes 300 µL-flow 3 mL/min–elution gradient ACN:H_2_O 50:50 going to 70:30–absorbance 650 and 600 nm). Pure fractions were combined, concentrated by rotary evaporation and the remaining H_2_O removed by lyophilization yielding a green powder **7** (30.5 mg, 35%) of greater than 98% purity, mp 35–40 °C. ^1^H NMR (400 MHz, CDCl_3_) δ: 8.08 (dd, *J* = 14.9, 7.8 Hz, 8H), 7.51–7.37 (m, 6H), 7.12–6.98 (m, 7H), 5.34 (br s, 1H), 4.58 (s, 2H), 3.90 (s, 3H), 3.71 (t, 2H), 3.67–3.53 (m, ~48H), 2.87 (br. s, 1H) ppm; ^19^F NMR (376 MHz, CDCl_3_) δ: −131.82 (q, *J* = 31.9 Hz) ppm. MALDI (Q-TOF): *m/z* calcd. for C_59_H_75_BF_2_N_4_NaO_15_^+^ [M + Na]^+^ 1151.51182; found 1151.3493.

*Pegylated fluorophore* (**8**). Compound **4** (22.0 mg, 0.032 mmol) and *O*-(2-aminoethyl) polyethylene glycol 4.6 kDa (163.8 mg, 0.035 mmol) were dissolved in anhydrous DMSO (2.0 mL) and stirred at rt for 1 h under a N_2_ atmosphere. The solvent was removed by lyophilization and the crude product dissolved in H_2_O (20 mL) and extracted with CH_2_Cl_2_ (3 x 20 mL). The organic layers were combined, washed with aqueous HCl (pH 5, 20 mL), H_2_O (20 mL) and brine (20 mL), dried over anhydrous Na_2_SO_4_, filtered and evaporated to dryness. The green solid was dissolved in H_2_O (1.5 mL) and was first purified by aqueous size exclusion chromatography (Sephadex G-25). The combined fractions were dried by lyophilization which yielded a green powder **8** (164.2 mg, 96% yield). For in vivo studies, this material was further purified as follows: diluted in ACN:H_2_O (3.0 mL; 47.5:52.5), filtered through a PTFE 0.45 µM syringe filter and purified by reverse phase semi-prep chromatography (YMC-Triart Phenyl, 10 × 150 mm I.D. S-5 µm, 12 nm; injection volumes 300 µL-flow 3 mL/min-elution gradient ACN:H_2_O 47.5:52.5 going to 70:30-absorbance 650 and 600 nm). Pure fractions were combined, concentrated by rotary evaporation and the remaining H_2_O removed by lyophilization yielding a green powder **8** (101.8 mg, 61%) of greater than 99% purity, mp 60–65 °C. ^1^H NMR (400 MHz, CDCl_3_) δ: 8.08 (dd, *J* = 7.6 Hz, 8H), 7.49–7.38 (m, 6H), 7.09–6.99 (m, 6H), 4.58 (s, 2H), 3.89 (s, 3H), 3.84–3.79 (t, 2H), 3.63 (m, ~475H), 3.48–3.43 (t, 2H) 2.82 (br. s., 1H, OH) ppm (NH not observed); ^19^F NMR (376 MHz, CDCl_3_) δ: -131.83 (q, *J* = 31.8 Hz) ppm. MALDI (Q-TOF): *m/z* calcd. for C_243_H_443_BF_2_N_4_NaO_107_^+^ [M + Na]^+^ 5201.9300; found 5207.8278.

*Pegylated fluorophore* (**9**). Compound **4** (9 mg, 0.013 mmol) and *O*-(2-aminoethyl) polyethylene glycol 9 kDa (140.3 mg, 0.016 mmol) were dissolved in anhydrous DMSO (1.0 mL) and stirred at rt for 1 h under an N_2_ atmosphere. The solvent was removed by lyophilization and the crude product dissolved in H_2_O (20 mL) and extracted with CH_2_Cl_2_ (3 × 20 mL). The organic layers were combined, washed with aqueous HCl (pH 5, 20 mL), H_2_O (20 mL), brine (20 mL), dried over anhydrous Na_2_SO_4_, filtered and evaporated to dryness. The green solid was dissolved in H_2_O (1.5 mL) and was first purified by aqueous size exclusion chromatography (Sephadex G-25). The combined fractions were dried by lyophilization which yielded a green powder **9** (119.4 mg, 95% yield). For in vivo studies, this material was further purified as follows: diluted in ACN:H_2_O (2.5 mL; 47.5:52.5), filtered through a PTFE 0.45 µM syringe filter and purified by reverse phase semi-prep chromatography (YMC-Triart Phenyl, 10 × 150 mm I.D. S-5 µm, 12 nm; injection volumes 300 µL-flow 3 mL/min-elution gradient ACN:H_2_O 47.5:52.5 going to 70:30-absorbance 650 and 600 nm). Pure fractions were combined, concentrated by rotary evaporation and the remaining H_2_O removed by lyophilization yielding a green powder **9** (84.4 mg, 68%) of greater than 96% purity, mp 65–70 °C. ^1^H NMR (400 MHz, CDCl_3_) δ: 8.08 (dd, *J* = 7.6 Hz, 8H), 7.45 (m, 6H), 7.10–7.00 (m, 6H), 5.34 (br. s., 1H, NH), 4.58 (s, 2H), 3.90 (s, 3H), 3.83–3.80 (t, 2H), 3.64 (m, ~1250H), 3.48–3.45 (t, 2H) 2.90 (br. s., 1H, OH) ppm; ^19^F NMR (376 MHz, CDCl_3_) δ: –131.83 (q, *J* = 31.4 Hz) ppm. MALDI (Q-TOF): *m/z* calcd. for C_441_H_839_BF_2_N_4_NaO_206_^+^ [M + Na]^+^ 9560.5252; found 9563.7042.

*Pegylated fluorophore* (**10**). Compound **6** (16.2 mg, 0.019 mmol) was dissolved in anhydrous DMSO (2.0 mL), *O*-(2-aminoethyl) polyethylene glycol 0.55 kDa (23.2 mg, 0.0425 mmol) was added and the reaction stirred for 1 h at rt under N_2_. The solvent was removed by lyophilization and the crude product was dissolved in H_2_O (10 mL) and extracted with CH_2_Cl_2_ (3 × 10 mL). The organic layers were combined, washed with aqueous HCl (pH 5, 10 mL), H_2_O (10 mL), brine (10 mL), dried over anhydrous Na_2_SO_4_, filtered and evaporated to dryness. The green solid was dissolved in H_2_O (2.0 mL) and was first purified by aqueous size exclusion chromatography (Sephadex G-25). The combined fractions were dried by lyophilization which yielded a green powder **10** (31.5 mg, 96% yield). For in vivo studies, this material was further purified as follows: diluted in ACN:H_2_O (0.5 mL; 50:50), filtered through a PTFE 0.45 µM syringe filter and purified by reverse phase semi-prep chromatography (YMC-Triart Phenyl, 10 × 150 mm I.D. S-5 µm, 12 nm; injection volumes 300 µL-flow 3 mL/min-elution gradient ACN:H_2_O 50:50 going to 70:30-absorbance 650 and 600 nm). Pure fractions were combined, concentrated by rotary evaporation and the remaining H_2_O removed by lyophilization yielding a green powder **10** (9.5 mg, 29%) of greater than 99% purity, mp 40–45 °C. ^1^H NMR (400 MHz, CDCl_3_): δ 8.08 (dd, *J* = 11.2, 8.1 Hz, 8H), 7.53–7.41 (m, 6H), 7.10–7.02 (m, 6H), 5.34 (d, 2H, NH), 4.59 (s, 4H), 3.71 (t, 4H), 3.67–3.53 (m, ~100H) ppm (OH not observed); ^19^F (376 MHz, CDCl_3_): δ –131.71 (q, *J* = 31.4 Hz) ppm. MALDI (Q-TOF): *m/z* calcd. for C_84_H_124_BF_2_N_5_NaO_28_^+^ [M + Na]^+^ 1722.8386; found 1725.3277.

*Pegylated fluorophore* (**11**). Compound **6** (19.5 mg, 0.023 mmol) was dissolved in anhydrous DMSO (2.0 mL), *O*-(2-aminoethyl) polyethylene glycol 2.9 kDa (144.7 mg, 0.05 mmol) was added and the reaction stirred for 1 h at rt under N_2_. The solvent was removed by lyophilization and the crude product was dissolved in H_2_O (20 mL) and extracted with CH_2_Cl_2_ (3 × 20 mL). The organic layers were combined, washed with aqueous HCl (pH 5, 20 mL), H_2_O (20 mL), brine (20 mL), dried over anhydrous Na_2_SO_4_, filtered and evaporated to dryness. The green solid was dissolved in H_2_O (1.0 mL) and was first purified by aqueous size exclusion chromatography (Sephadex G-25). The combined fractions were dried by lyophilization which yielded a green powder **11** (140.2 mg, 94% yield). For in vivo studies, this material was further purified as follows: dissolved in ACN:H_2_O (1.5 mL; 45:55), filtered through a PTFE 0.45 µM syringe filter and purified by reverse phase semi-prep chromatography (YMC-Triart Phenyl, 10 × 150 mm I.D. S-5 µm, 12 nm; injection volumes 300 µL–flow 3 mL/min–elution gradient ACN:H_2_O 45:55 going to 70:30–absorbance 650 and 600 nm). Pure fractions were combined, concentrated by rotary evaporation and the remaining H_2_O removed by lyophilization yielding a green powder **11** (78.5 mg, 53%) of greater than 99% purity, mp 50–55 °C. ^1^H NMR (400 MHz, CDCl_3_): δ 8.12–8.02 (m, 8H), 7.44 (m, 6H), 7.10–7.01 (m, 6H), 4.58 (s, 4H), 3.83–3.78 (t, 4H), 3.72–3.54 (m, ~680 H), 3.47–3.43 (m, 4H) ppm (NH or OH not observed); ^19^F NMR (376 MHz, CDCl_3_): δ –131.71 (q, *J* = 31.5 Hz) ppm. MALDI (Q-TOF): *m/z* calcd. for C_296_H_549_BF_2_N_5_NaO_134_^+^ [M + Na]^+^ 6390.6252; found 6394.0038.

*Pegylated fluorophore* (**12**). Compound **6** (10.1 mg, 0.012 mmol) was dissolved in anhydrous DMSO (1.0 mL), *O*-(2-aminoethyl) polyethylene glycol 4.6 kDa (122.4 mg, 0.027 mmol) was added and the reaction stirred for 1 h at rt under N_2_. The solvent was removed by lyophilization and the crude product was dissolved in H_2_O (10 mL) and extracted with CH_2_Cl_2_ (3 x 10 mL). The organic layers were combined, washed with aqueous HCl (pH 5, 10 mL), H_2_O (10 mL), brine (10 mL), dried over anhydrous Na_2_SO_4_, filtered and evaporated to dryness. The green solid was dissolved in H_2_O (1.0 mL) and was first purified by aqueous size exclusion chromatography (Sephadex G-25). The combined fractions were dried by lyophilization which yielded a green powder **12** (112.1 mg, 96% yield). For in vivo studies, this material was further purified as follows: dissolved in ACN:H_2_O (1.25 mL; 45:55), filtered through a PTFE 0.45 µM syringe filter and purified by reverse phase semi-prep chromatography (YMC-Triart Phenyl, 10 × 150 mm I.D. S-5 µm, 12 nm; injection volumes 300 µL- ACN:H_2_O 45:55 going to 70:30-flow 3 mL/min–absorbance 650 and 600 nm). Pure fractions were combined, concentrated by rotary evaporation and the remaining H_2_O removed by lyophilization yielding a green powder **12** (67.0 mg, 57%) of greater than 96% purity, mp 65–70 °C. ^1^H NMR (400 MHz, CDCl_3_): δ 8.11–8.03 (m, 8H), 7.50–7.40 (m, 6H), 7.10–7.01 (m, 6H), 5.34 (br. s., 2H, NH) 4.58 (s, 4H), 4.12 (br. s., 2H, OH) 3.80 (t, 4H), 3.74–3.53 (m, ~900H), 3.46 (t, 4H) ppm; ^19^F NMR (376 MHz, CDCl_3_): −131.72 (q) ppm. MALDI (Q-TOF): *m*/*z* calcd. for C_452_H_861_BF_2_N_5_NaO_212_^+^ [M + Na]^+^ 9824.6700; found 9827.7984.

### 3.3. Quantum yield Determination of Fluorophores *(**7**)–(**12**)*

Quantum yields were calculated using the comparative method and measured against the known standard **1a** in acetonitrile [28]. A triplicate of five absorption/emission spectra were recorded respectively for each fluorophore. Graphs of absorbance at excitation wavelength versus integrated fluorescence were generated to determine each quantum yield with correction for solvent refractive index included for water values.

### 3.4. Rodent Imaging Study

Compliance with ethical standards: Animal experiments conformed to guidelines from Directive 2010/63/EU of the European Parliament on the protection of animals used for scientific purposes. Experiments were licensed and approved by the Health Products Regulatory Authority Ireland (HPRA) project authorization number AE19127/P062. Protocols were also reviewed by the Royal College of Surgeons Animal Research Ethics Committee (AREC). Power calculations were reviewed and approved by the AREC biostatistician. Power calculations: Study cohort N number was calculated using the formula N = (Zα + Zβ)2 * (2σ2/δ2). σ (sigma) is the common variance of fluorescence being measured. δ (delta) is difference between background fluorescence and administered fluorophore. This difference is judged to be different that testing should generate a significant result (δ = (µ2 − µ1)). Based on prior unpublished data from the group, we determined that δ = 0.72 × 10^8^ p/s/cm^2^/str, σ= 0.38 × 10^8^ p/sec/cm^2^/str and using standard normal distribution tables α = 0.01667 (modified by Bonferoni correction for multiple testing and β = 0.8. Therefore, a study cohort size of N = 5.817 ≈ 6 animals per group was used. Animals (6–8 weeks 150–200 g, male/female 50:50 ratio) were purchased from Charles River (Canterbury, UK) and maintained in the RCSI animal facility.

### 3.5. In Vivo Rodent Screening of Fluorophores ***8***, ***9***, ***11***, ***12***

N = 24 Sprague–Dawley (CRUK) rats aged 6–8 weeks old were divided into 4 groups with each group containing 6 rats. Rats were anaesthetized with 4% isoflurane and 0.8 L/min O_2_ and maintained at 2% isoflurane and 0.8 L/min O_2_ for the duration of the experiment. Each group of 6 rats were injected via the lateral tail vein with one of the fluorophores **8**, **9**, **11**, **12** at 2 mg/kg, and were imaged/videoed for 90 min under the Quest imaging platform with excitation at 685 nm and collection between 700–800 nm. A signal was detected in the lower abdomen/bladder of animals. After the imaging session was complete, the animals were euthanized by anaesthetic overdose followed by cervical dislocation.

### 3.6. Dose De-Escalation Study for Fluorophore ***12***

N = 6 Sprague–Dawley rats aged 6–8 weeks old were anaesthetized with 4% isoflurane and 0.8 L/min O_2_ and maintained at 2% isoflurane and 0.8 L/min O_2_ for the duration of the experiment. Animals were injected with a dose of 0.2 mg/kg in the lateral tail vein, and were imaged as before for 90 min. After the imaging session was complete, the animals recovered in a warmed recovery cage until they regained full consciousness. One week later, the same group of animals underwent the same imaging parameters but the dose of fluorophore was reduced. This de-escalation of dose will occurred 2 further times with the doses dropping to 0.1 mg/kg and 0.05 mg/kg with a week’s recovery between each imaging session. After the last imaging session, rats were euthanized by anaesthetic overdose followed by cervical dislocation.

### 3.7. Ureter Identification in Rodent Model Using ***12***

N = 3 female Sprague–Dawley rats aged 6-8 weeks old were anaesthetized as above. Animals were placed on a pre-calibrated heating pad in a sterile surgical environment. The hair from the abdomen was removed using depilatory cream. The surgical site was swabbed with alcohol and povidone iodine mixture. A midline incision of 3 cm and a left side incision was performed on the abdomen to expose the bladder, the left/right ureter, kidney, liver and spleen. The animals were injected via the lateral tail vein with 0.5 mg/kg of **12**, and then were imaged as before. Rats were imaged for 90 min. After the imaging session, animals were euthanized by anaesthetic overdose cervical dislocation.

### 3.8. Toxicity Study of ***12*** in Rodents

This study was independently carried out in the test facility of Citoxlab France (subsidiary of Charles River), Evreux, France. The study was performed in a test facility certified by the French National Authorities for Good Laboratory Practice compliance and followed established practices and standard operating procedures of Citoxlab France. The study was conducted in compliance with animal health regulations, in particular: Council Directive No. 2010/63/EU and French decree No. 2013-118 on the protection of animals used for scientific purposes. The Citoxlab France Ethics Committee reviewed and approved the study plan in order to assess compliance as defined in Directive 2010/63/EU and in French decree No. 2013-118.

The objective of this study was to evaluate the potential toxicity of **12**, following (i) 7 days of treatment of 2 mg/kg daily dose and (ii) a single dose of 10 mg/kg by intravenous route (bolus). Nine male rats (Sprague–Dawley) sourced from Charles River Laboratories Italia, Calco, Italy were used for the study. At the beginning of the treatment period, the animals were 6 weeks old and weighed between 130 and 240 g. Upon arrival at Citoxlab France, the animals were given a clinical examination to ensure good condition and were acclimated to the study conditions for 5 days before the beginning of the treatment period. Animals were housed in a secure rodent unit with conditions set as follows: temperature: 22 ± 2°C, relative humidity: 50 ± 20%, light/dark cycle: 12 h/12 h, ventilation: 8 to 15 cycles/hour of filtered, non-recycled air. The animals were housed in groups of 3, in polycarbonate cages with stainless steel lids containing autoclaved sawdust. Each cage contained at least two objects for environmental enrichment. All animals had free access to SSNIFF rat/mouse pelleted maintenance diet and to tap water (filtered with a 0.22 µm filter) contained in bottles.

A PBS solution of **12** (2 mg/kg) was administered by intravenous route (bolus), with a constant dosage-volume of 5 mL/kg used for the treated group (*n* = 3). Each animal received this dosage daily for seven consecutive days. In the same manner, a single 10 mg/kg dosage was used to treat another group (*n* = 3). The quantity of dose administered to each animal was adjusted according to the most recently recorded body weight. The control animal group were administered PBS by intravenous route (bolus) for seven consecutive days (*n* = 3). Prior to blood sampling, the animals were deprived of food for an overnight period of 14 h. Blood samples were taken from the orbital sinus of the animals under light isoflurane anesthesia, into appropriate tubes.

### 3.9. Porcine Imaging Study

The present study is part of the ELIOS protocol (endoscopic luminescent imaging for oncology surgery), fully approved by the local Ethical Committee on Animal Experimentation (ICOMETH No. 38.2016.01.085), and by the French Ministry of Superior Education and Research (MESR) (APAFIS#87212017013010316298-v2). Six adult female pigs (Sus scrofa domesticus, mean weight: 9.6 ± 1.3 kg) were used. Animals were managed according to the directives of the European Community Council (2010/63/EU) and ARRIVE guidelines [38]. Following an acclimatation period in our animal keeping facility, the animals received premedication by means of an intramuscular injection of Zolazepam and Tiletamine 10 mg/kg (Zoletil ND, Virbac, Carros, France). Anesthesia induction was achieved through an intravenous injection of Propofol 3 mg/kg (Propofol Lipuro ND, B Braun, Sarlat, France) together with rocuronium 0.8 mg/kg (Esmeron ND, MSD, France). After intubation, the pigs were mechanically ventilated throughout the experiment and were sedated via an inhalation of isoflurane 2–3% (Isoflurin ND, Axience, Pantin, France). Three experimental doses of **12** as 0.5, 0.25 and 0.1 mg/kg (*n* = 2 for each dose) were administered by intravenous route (bolus) with imaging for 120 min. During the experiment, analgesia was ensured with intramuscular buprenorphine (Buprecare ND, Axience, Pantin, France) 0.01 mg/kg. At the end of the experimental procedure, pigs were sacrificed under deep anesthesia (Isoflurane 5%) with a lethal intravenous application of Pentobarbital 40 mg/kg (Exagon ND, Axience, Pantin, France).

## 4. Conclusions

Mono- and bis-PEGylated fluorophores **7**–**12** were synthesized in seven steps with greater than 95% purity, with each having similar emission spectra spanning the 700–800 nm wavelength region. Fluorophore bis-PEGylation resulted in higher aqueous fluorescence quantum yields with PEG chain lengths of 2.9 to 4.6 kDa of **11** and **12** being optimal. The advantage of substituting with two PEG groups was also seen in an in vivo rodent model screen of two mono- and two bis-substituted derivatives from which fluorophore **12** was selected for more in-depth studies. An *in rodent* dose de-escalation study with **12**, showed that its emission could be detected from urine in the bladder at doses as low as 0.05 mg/kg. Rodent toxicity studies indicated **12** as a safe surgical tool with NOAEL established as 10 mg/kg/day. It was successfully employed for fluorescence ureter identification with the preference for renal excretion notable through comparative fluorescence intensities from the ureters, kidneys and liver. Ureteral identification with **12** was successfully performed in a larger animal porcine model under abdominal surgical conditions. Three tested doses of 0.5, 0.25 and 0.1 mg/kg all successfully identified fluorescent ureters within 20 min of administration and were sustained for 120 min. The use of 3-D emission heat map imaging allowed the spatial and temporal changes in intensity due to the peristaltic waves of urine being transferred from kidneys to the bladder to be clearly distinguished from surrounding tissues. As the emission of **12** was spectrally separable from the clinically used ICG it is envisaged that their combined use could be a step towards intraoperative colour coding of different tissues.

## Data Availability

Image files available upon request from the corresponding author.

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
