# Peer review of "BF2-Azadipyrromethene Fluorophores for Intraoperative Vital Structure Identification"

_molecules, 2023, doi:10.3390/molecules28052167_

Round 1

Reviewer 1 Report

The manuscript submitted by Donal F. O’Shea and co-workers is of very high interest for Molecules, the work is properly carried out and deserves to be published, data support conclusions. The work is very useful as there is a lack of NIR fluorophores in clinics, and in vivo results are impressive. Just replace tert-butyl ester saponification by cleavage as saponification is carried out in basic media.

Author Response

Thanks for the constructive feedback on our manuscript. 

The term saponification in the text has been changed to cleavage. 

Reviewer 2 Report

In this work, Peng et al. reported the design, synthesis, and in vivo imaging studies of a series of mono- and bis-polyethylene glycol (PEG) substituted BF2-azadipyrromethene fluorophores. The compounds are well characterized and the research is conducted correctly. This is a very good article addressing an important issue and is very well presented. Thus, it would be an exciting addition to the related area and warrants publication in Molecules after addressing some of the questions and comments outlined below:

1. In the photophysical properties section, please provide the details for the quantum yield determination of 7-12.

2. The authors should carefully read the grammatical errors and inappropriate scientific language throughout the text.

For instance, on Page 7, the sentence ‘…As would be expected fluorescence scattering was noted for...’ has a grammatical issue.

3. Conclusion is not appropriately given. The major findings and significance should be summarized.

Author Response

Thanks for the constructive feedback on our manuscript. The following changes have been made

  1. Additional text has been added to the experimental section describing quantum yield measurements on manuscript page 18.
  2. The sentence has been rewritten to resolve the grammar issue.
  3. The conclusion is at section 4 on page 20 of the manuscript.

Reviewer 3 Report

This article presents the synthesis, the optical properties studies and the application of azabodipy compounds bearing PEGs of various sizes. Half of these compounds are bearing one PEG chain, while the others two chains.

The synthesis and the purifications are well described and the characterization is very convincing. A significant effort has been made to purify Pegylated compounds, using advanced techniques. The optical properties were done properly. The comparison of the fluorescence in acetonitrile and water is very indicative of the benefit of having two PEG chains by comparison with 1 PEG chain. An intriguing result is that the fluorescence quantum yield is more related to the presence of two chains than the overall mass of PEG.

A good point is that, prior in vivo application, the fluorescence of the compounds was tested in capillaries in intralipid solution.

Then fluorophores were tested with rodents which permitted to investigate their accumulation, and their toxicity.

In the end a porcine model was tested in presence ICG to demonstrate that these compounds can be successfully applied in surgery.

 I strongly recommend the publication of this article.

Author Response

Thanks for the constructive feedback and positive comments for our manuscript.